# The Impacts of Self-Construal and Perceived Risk on Technology Readiness

Jaeun Choi [1] and Dongho Yoo [2,*]

1    Regional Tourism Research Division, Korea Culture & Tourism Institute, Seoul 157857, Korea;
jechoi@kcti.re.kr
2    School of Global Business, Halla University, Wonju 26404, Korea
*    Correspondence: dongho.yoo@halla.ac.kr; Tel.: +82-33-760-1414

**Abstract:** This paper explores how different self-construals interact with perceived risk and influence tourism consumers' technology readiness toward tourism mobile apps. The study used 284 tourist participants to show that tourism consumers, regardless of self-construal type, have a negative perception of technology readiness when they perceive a high level of risk. Furthermore, those with independent self-construal have a more positive perception of technology readiness than those with an interdependent self-construal when they perceive a low level of risk. The results also show that technology readiness mediates the relationship between self-construal and use intention toward mobile apps. Based on these results, we proposed the following practical implications. First, companies need to find and lower potential risks that can be perceived in tourism mobile apps. They must also deliver different messages according to consumers' self-construal. Companies should provide independents with information related to the positive benefits obtained through the mobile app, and interdependents should be provided with information that reduces perceived losses.

**Keywords:** self-construal; perceived risk; technology readiness; tourism consumer; mobile app



## 1. Introduction

Smartphones have evolved to become minicomputers that support a variety of information services, often in the form of mobile apps that are available to use anytime and anywhere [1,2]. The functionality of such information services plays an important role in tourism consumers' decision-making. Indeed, prior research indicates that the readily available information on smartphone mobile apps assists tourism consumers [2–6]. In particular, Wang, Park and Fesenmaier [2] found that mobile apps enable tourism consumers to cope with unexpected events and to engage in activities more effectively and efficiently. Thus, tourism consumers can enhance the quality of travel through smartphone mobile apps [5,7].

Although mobile apps have positive effects on tourism consumers' decision-making, many tourism consumers do not use them. In order to determine whether tourists use mobile apps on their trips, we ran a simple survey. The survey respondents ($n$ = 100) were American tourists in Korea. The result showed that 55% had used a mobile app and 45% had not ($p$ > 0.1). Thus, approximately half of the respondents had not used a mobile app. This result is similar to that of the Expedia/Egencia survey in which 58% of respondents reported that they had used a mobile app [8]. Furthermore, when we asked our survey participants about the type of searching tool that they mainly used during their trip to Korea, 62% said that they had used the mobile Web and 13% said that they had used a mobile app ($p$ < 0.001). In particular, this result showed that none of the respondents had used the tourism mobile app developed by the Korea Tourism Board. This suggests that although tourism mobile apps are developed in order to attract tourists, tourists do not use them. To the best of our knowledge, no research has explored this issue. Consequently, we investigate this important topic.

According to technology readiness of Parasuraman [9], when people embrace and use new technology, they focus not only on positive aspects such as optimism and innovativeness but also on negative aspects such as discomfort and insecurity. Consumers' technology readiness affects their perception of technology (perceived usefulness and perceived ease of use) and intention to use the technology. However, although technology readiness is an essential variable for consumers' acceptance of technology, previous studies tended to focus on the consequences of technology readiness [10]. In other words, there was limited research on antecedents of technology readiness. This situation raises two questions. First, what type of tourism consumers focus on positive rather than negative aspects? Second, when do tourism consumers focus on negative rather than positive aspects?

This research approaches these questions from the perspective of self-construal and perceived risk in the context of mobile apps. We propose that tourism consumers' technology readiness for mobile apps differs according to self-construal. Specifically, tourism consumers with independent self-construal, which focuses on gains, place relatively more weight on the positive aspects of mobile apps. In contrast, tourism consumers with interdependent self-construal, which focuses on losses, place relatively more weight on the negative aspects of mobile apps. However, we also propose that the relationship between self-construal and technology readiness is qualified by perceived risk. If perceived risk is high, tourism consumers place relatively more weight on negative aspects and less weight on positive aspects regardless of self-construal because a high level of perceived risk prompts thoughts of loss.

## 2. Theoretical Background

### 2.1. Technology Readiness

As tourism consumers' search for tourism information through smartphones increases, various studies investigated the use of mobile apps by these consumers. Previous studies show that various functions of tourism mobile apps, such as mobile travel guides, recommendation systems, location-based services, and directions, are suitable to satisfy the needs of tourism consumers [11,12]. As a result, mobile apps are being used as a marketing tool to attract consumers in the tourism industry [13,14]. In addition, mobile apps increase consumers' negative perceptions, such as privacy and security concerns [15]. Therefore, to investigate consumers' acceptance intentions for mobile apps, it is necessary to examine both consumers' positive and negative perceptions about technology. Technology readiness is a theory suitable for examining tourism consumers' conflicting perceptions.

Technology readiness is defined as "people's propensity to embrace and use new technologies for accomplishing goals in home life and at work" (Parasuraman, 2000, p. 308). The technology readiness construct is composed of four dimensions: optimism, innovativeness, discomfort, and insecurity [9,16]. Optimism and innovativeness are drivers of technology readiness, and discomfort and insecurity are inhibitors.

Optimism refers to a belief that new technologies offer people greater control, flexibility, and efficiency in their lives. Optimistic consumers tend to focus more on positive than negative consequences [17]; thus, highly optimistic consumers are more likely to use new technologies and to try to use the innovative functions of these technologies [18]. The second driver of technology readiness, innovativeness, refers to the tendency of people to be pioneers or thought leaders and to try out new things [9]. Thus, innovativeness has a positive influence on usage variety [19].

Discomfort refers to a perception about technology's uncontrollability and a feeling of being overcome by technology. Consumers with high discomfort levels experience fear based on lack of comprehension and the learning costs of technology-based products and services [20]. They regard technology as complex and often have negative feelings about it that include aggravation, frustration, and disappointment [21]. As a result, such consumers use new technology-based products less frequently. The other technology readiness inhibitor, insecurity, refers to a distrust of technology and the associated technical ability required for technology to work properly. Consumers with high insecurity levels

are skeptical about new technologies [18]; as a result, they have less interest in accepting and using new technologies [22].

Technology readiness plays a key role in consumer acceptance of technology in service areas such as tourism. For example, Pradhan et al. [23] found that consumers with high technology readiness have more positive attitudes toward smart-devices because they perceive more benefits than losses. Prodanova, San-Martín and Jimenez [5] showed that the higher the level of technology readiness of consumers, the more positive attitudes they held toward mobile advertising related to tourism and the more positive purchase intention. Lin et al. [24], who investigated the relationship between technology readiness and technology acceptance model, found that technology readiness enhances consumers' perceived usefulness and ease of use. As a result, it positively affects consumers' intention to use technology. In other words, technology readiness is a meaningful variable in examining consumer acceptance of technology. However, compared to the high interest in the consequences of technology readiness on consumers' intentions and behaviors, research on the antecedents of technology readiness was limited [10]. In addition, prior studies examining the antecedents of technology readiness mainly suggest demographic characteristics, such as age and education level, as being key variables. It was not possible to reveal what disposal characteristics and consumer traits influence technology readiness. Therefore, this was examined through self-construal, which embodies consumers' individual characteristics.

### 2.2. Self-Construal

Self-construal refers to an individual's view of self and the self-schema structure [25–27] and reflects the relation between self and social interaction with others [28]. Individuals with an independent self-construal (independents) tend to primarily think and behave with reference to their internal thoughts. Thus, independents place a high value on autonomy and self-reliance. Conversely, individuals with an interdependent self-construal (interdependents) tend to primarily think and behave with reference to the thoughts of others. Thus, interdependents place a high value on interpersonal harmony and relationships [27,29]. Consequently, self-construal influences individual judgments and behaviors. For example, Chen [30] found that independents place more weight on internal reference price than external reference price, whereas interdependents place more weight on external reference price than internal reference price. In addition, according to Simpson et al. [31] who examined the influence of self-construal and public recognitions on consumers' intention to donate, independents have more positive donation intentions in private condition than public condition, while interdependents have more positive donation intentions in public condition than private condition.

Early research on self-construal found that individual self-construal varies according to culture. Specifically, North Americans and other Westerners have independent self-construal, while South and East Asians have interdependent self-construal [27,32,33]. However, subsequent research discovered that regardless of culture, independent and interdependent self-construals coexist within an individual [34–36].

Self-construal influences individual goals and the way that people regard objects. Independents' goals are to achieve and succeed relative to others. In contrast, interdependents' goals are to belong to groups and fulfill duties and responsibilities to others [37]. Further, empirical research has shown that different types of self-construal influence individual decision-making strategies. While independents tend to use an approach strategy that focuses on the positive aspects of options, interdependents tend to use an avoidance strategy that focuses on the negative aspects of options [38–41]. Thus, independents want to achieve a positive end state and are sensitive to gains/non-gains, whereas interdependents want to achieve a non-negative end state and are sensitive to loss/non-loss [29,42].

Consequently, technology readiness for mobile apps differs according to individual self-construal. Specifically, independents (who are sensitive about gains) are more likely to focus on the positive aspect of technology, while interdependents (who are sensitive about losses) are more likely to focus on the negative aspect of technology. Thus, we propose the following hypothesis.

**Hypothesis 1 (H1).** *Compared with interdependents, independents have a positive perception about their technology readiness for mobile apps.*

*2.3. Perceived Risk*

Prior research in social science has noted the important role of perceived risk when evaluating consumer behaviors [43–46]. Such perceived risk can be considered in the context of uncertainty and consequences: the higher the uncertainty level and/or the greater chance of associated negative consequences, the higher the perceived risk [47,48]. Thus, perceived risk is defined as uncertainty, which means that negative utility is expected in the use and purchase of products and services.

Risk perceived by the consumer can be identified and measured by referring to several sources [46]. For example, Kaplan et al. [49] suggest that sources of risk include financial, performance, social, physical, and psychological and that one or more of these drives an individual's overall perceived risk [50]. Thus, consumers' perceived risk is an overall risk that derives from just one source or a combination of sources. Consequently, perceived risk influences individual behaviors, and a high level of perceived risk leads to less alternative search and word of mouth (WOM) communication by individuals [51].

Further, individuals' perceived high risk focuses on the negative aspects of a situation and places more weight on avoiding the risk [52]. Lin, Chang and Lin [52] showed that consumers are sensitive to losses in high-risk situations regardless of their self-construal. In addition, Lin, Chang and Lin [52] found that a high level of perceived risk increases consumers' prevention orientation in order to avoid losses. In a similar vein, Lee and Aaker [53] showed that negative information is more persuasive than positive information when consumers perceive high risk.

Thus, individuals tend to focus on negative aspects rather than positive aspects regardless of their self-construal when a situation is perceived to be high risk. Consequently, we hypothesize that individual self-construal does not influence technology readiness for mobile apps in a situation of high perceived risk. However, independents have a more positive perception of technology readiness for mobile apps compared with interdependents in a situation of low perceived risk.

**Hypothesis 2 (H2).** *Compared with interdependents, independents have a more positive perception of technology readiness for mobile apps in a situation of low perceived risk. However, this perception does not apply in a situation of high risk.*

**3. Materials and Methods**

*3.1. Development of Stimulus Materials*

A pretest (*n* = 33) was run to develop the study materials. Participants randomly read one of two descriptions of a tourism mobile app that was either high or low risk. Specifically, the participants with the high-risk mobile app read about the provision of GPS services without approval, accommodation search and reservation services with payment through a mobile app, the uploading of advertisements for unknown companies, and the failure to implement security programs. In contrast, the participants with the low-risk mobile app read about the provision of GPS services with approval, accommodation search and reservation services, the uploading of advertisements for well-known companies, and the implementation of security programs. The participants were then asked to indicate their perceptions of risk on 7-point scales. In this context, perceived risk was measured by 10 items extracted from 24 listed by Featherman and Pavlou [54].

As expected, the participants perceived greater risk in high-risk conditions (M = 4.31, SD = 1.21) than in low-risk conditions (M = 3.27, SD = 0.70; t = 2.964, *p* < 0.01).

### 3.2. Research Procedure

Two hundred and eighty-four participants who were currently traveling or planning to travel were recruited. They had varying cultural backgrounds (American and Korean) to ensure generalizability and were randomly assigned to two groups (high risk/low risk). In the case of Americans, surveys were conducted at Incheon International Airport, targeting travelers entering Korea. In the case of Koreans, a Korean online survey company was used to conduct surveys targeting people planning to travel. People currently traveling or planning to travel were selected as subjects to increase their commitment to this survey related to using the tourism mobile app. Table 1 shows their characteristics.

**Table 1.** Sample characteristics.

|  | Characteristics | *n* | % |
|---|---|---|---|
| Country | US | 149 | 52.5 |
|  | Korean | 135 | 47.5 |
| Gender | Male | 140 | 49.3 |
|  | Female | 144 | 50.7 |
| Age | 10–19 | 5 | 1.8 |
|  | 20–29 | 149 | 52.5 |
|  | 30–39 | 50 | 17.6 |
|  | 40–49 | 33 | 11.6 |
|  | 50–59 | 26 | 9.2 |
|  | $\geq$60 | 21 | 734 |

The participants were asked to complete two unrelated tasks. In the first task, we measured participants' self-construal using Singelis [55] 24 items where 12 items = an independent subscale and 12 items = an interdependent subscale, together with a 7-point rating where 1 = very strongly disagree and 7 = very strongly agree. The participants' ratings on the two subscales were averaged to form an independent self-index (M = 4.73, SD = 0.63; $\alpha$ = 0.65) and an interdependent self-index (M = 4.17, SD = 1.03; $\alpha$ = 0.87). In accordance with prior findings [36,55,56], the correlation between the independent and interdependent self-indices was not significant (r = $-0.05$, $p > 0.1$). This result means that independent and interdependent self-construals are largely orthogonal.

After completing Singelis [55] scale, participants were presented with a description of a fictitious tourism mobile app (see Appendix A). The mobile app was developed directly by us and was created based on the Korea Tourism Organization's mobile app called "Visit Korea 3.0." Participants were presented with an image of the main screen where they could see the mobile app's functions. The description contained the perceived risk manipulation that was developed from the pretest. Participants answered questions about perceived risk, technology readiness, and use intention toward the tourism mobile app. The success of the perceived risk manipulation was measured with Belanche et al.'s [57] 3-item method using 7-point scales ($\alpha$ = 0.749). Technology readiness was measured by Jin's [58] 13-item method using 7-point scales where 7 items = the drivers' subscale ($\alpha$ = 0.883) and 6 items = the inhibitors' subscale ($\alpha$ = 0.814). The approach was modified from Parasuraman's [9] scale for a specific technology. Finally, use intention was measured with Jin's [58] 3-item method using 7-point scales ($\alpha$ = 0.792). All measurement items can be seen in Table 2.

After the experiment, we thanked the participants and gave each of them a souvenir that cost approximately USD 2.

**Table 2.** Measurement items.

| Variables | Items | Cronbach's Alpha |
|---|---|---|
| **Self-Construal** | | |
| Independent | "My personal identity, independent of others, is very important to me." | 0.751 |
| | "I enjoy being unique and different from others." | |
| | "Being able to take care of myself is a primary concern for me." | |
| | "Speaking up in a work/task group/class is not a problem for me." | |
| | "Having a lively imagination is important to me." | |
| | "I'd rather say "no" directly than risk being misunderstood." | |
| | "I am comfortable being singled out for praise or rewards." | |
| | "I am the same person at home that I am at school." | |
| | "I act the same way no matter who I am with." | |
| | "I feel comfortable using someone's first name soon after I meet them, even when they are much older than I am." | |
| | "I prefer to be direct and forthright when dealing with people I've just met." | |
| | "I value being in good health above everything." | |
| Interdependent | "My relationships with those in my group are more important than my personal accomplishments." | 0.870 |
| | "My happiness depends on the happiness of those in my group." | |
| | "I am careful to maintain harmony in my group." | |
| | "I would sacrifice my self-interests for the benefit of my group." | |
| | "I will stay in a group if they need me, even if I'm not happy with the group." | |
| | "I should take into consideration my parents' advice when making education and career plans." | |
| | "I respect decisions made by my group." | |
| | "If my brother or sisters fails, I feel responsible." | |
| | "I have respect for the authority figures with whom I interact." | |
| | "I would offer my seat on the bus to my professor." | |
| | "I respect people who are modest about themselves." | |
| | "Even when I strongly disagree with group members, I avoid an argument." | |
| **Technology Readiness** | | |
| Drivers | "Technological functions and services provided by mobile apps gives people more control over their daily lives." | 0.883 |
| | "You prefer to use the most advanced technological functions and services provided by mobile apps available." | |
| | "Technological functions and services provided by mobile apps make you more efficient in your occupation." | |
| | "Learning about technology can be as rewarding as the technology itself." | |
| | "You keep up with the latest technological developments and the advanced services provided by mobile apps in your areas of interest." | |
| | "You enjoy the challenge of figuring out high-tech gadgets and new issues." | |
| | "You find you have fewer problems than other people in making technology work and advanced function of mobile apps for you." | |
| Inhibitors | "Sometimes, you think that technology systems, functions and services are not designed for use by ordinary people." | 0.814 |
| | "Many new technological functions and services provided by mobile apps have health or safety risks that are not discovered until after people have used them." | |
| | "Technology and advanced services provided by mobile apps always seems to fail at the worst possible time." | |
| | "You do not feel confident doing business with a place that can only be reached mobile apps." | |
| | "You do not consider it safe to do any kind of financial business mobile app." | |
| | "The human touch is very important when doing business with a company." | |
| Perceived risk | "Using this mobile tour app causes me to be concerned with experiencing some kind of loss in the future." | 0.749 |
| | "I am vulnerable to the actions conducted by this mobile tour app." | |
| | "The actions conducted by this mobile tour app may cause problems and uncertain consequences for me. " | |
| Use intention | "I have a strong tendency to continuously use this mobile tour app." | 0.792 |
| | "I would recommended this mobile tour app to my friends" | |
| | "I would intend to get information from this mobile tour app." | |

## 4. Results

### 4.1. Manipulation Check

We analyzed the data using SPSS 21.0. A one-way analysis of variance (ANOVA) test on the perceived risk revealed a significant effect on risk level (F = 10.295, $p < 0.01$). Participants

who read the description of the high-risk mobile app had more to say about risk (M = 3.75) compared to those who read the description of the low-risk mobile app (M = 3.30).

### 4.2. Technology Readiness

Table 3 presents the correlations for all variables. A technology readiness index was calculated by subtracting the inhibitors from the drivers. Since independent self-construal and interdependent self-construal were measured as continuous variables rather than categorical variables, a regression analysis was performed, instead of an analysis of variance. In addition, to eliminate the possibility of multicollinearity, the two self-construal values were mean-centered. To test our hypotheses, we regressed the technology readiness index on participants' mean-centered independent and interdependent self-indices; the perceived risk level (coded as high risk = 1 and low risk = 0); and the two interaction terms of perceived risk level × independent self and perceived risk level × interdependent self (see Table 4). The overall model was significant ($R^2$ = 0.278, F (5, 278) = 22.846, $p < 0.001$). Furthermore, the main effects on risk level, independent self-index, and interdependent self-index were significant. The results showed the positive effect of the independent self-index ($\beta$ = 0.404, $p < 0.001$) together with the negative effects of the interdependent self-index ($\beta$ = −0.366, $p < 0.001$) and the perceived risk level ($\beta$ = −0.406, $p < 0.001$) on technology readiness. Thus, the first hypothesis is supported. More importantly, the interaction effects between perceived risk level and independent self-index ($\beta$ = −0.303, $p < 0.001$), and the interaction effect between perceived risk level and interdependent self-index ($\beta$ = 0.289, $p < 0.001$), were significant.

**Table 3.** Correlations.

| Variables | 1 | 2 | 3 | 4 |
|---|---|---|---|---|
| Independent self-construal | | | | |
| Interdependent self-construal | −0.045 | | | |
| Perceived risk level | 0.058 | −0.093 | | |
| Technology readiness | 0.151 * | −0.103 | −0.389 ** | |
| Use intention | 0.199 ** | −0.024 | −0.205 ** | 0.556 ** |

\* $p < 0.05$, \*\* $p < 0.01$.

**Table 4.** Results of technology readiness regression analysis on different self-construals and perceived risk level.

| Model | Unstandardized Coefficients | | Standardized Coefficients | t | p |
|---|---|---|---|---|---|
| | Beta | Standard Error | Beta | | |
| (Constant) | 0.513 | 0.137 | 0 | 3.753 | 0.000 |
| Independent self-construal (A) | 1.231 | 0.213 | 0.404 | 5.767 | 0.000 |
| Interdependent self-construal (B) | −0.685 | 0.135 | −0.366 | −5.082 | 0.000 |
| Perceived risk level (C) | −1.564 | 0.196 | −0.406 | −7.977 | 0.000 |
| A * C | −1.347 | 0.312 | −0.303 | −4.315 | 0.000 |
| B * C | 0.764 | 0.192 | 0.289 | 3.984 | 0.000 |

To achieve a better understanding of the interaction effects, we examined the effects of the two self-construal indices on participants' technology readiness toward the mobile apps at each perceived risk level. First, for the high perceived risk level, the independent self-index ($\beta$ = −0.046, $p > 0.1$) and interdependent self-index ($\beta$ = 0.049, $p > 0.1$) were not significant. For the low perceived risk level, the independent self-index was positive and significant ($\beta$ = 0.372, $p < 0.001$), whereas the interdependent self-index was negative and significant ($\beta$ = −0.328, $p < 0.001$). These results provide evidence that independent self-construal had a positive influence on technology readiness toward the mobile app with the low perceived risk level but had no effect on technology readiness toward the mobile app with the high perceived risk level. However, interdependent self-construal had a negative influence on technology readiness toward the mobile app with the low perceived risk level but had no effect on technology readiness toward the mobile app with the high perceived risk level. Thus, the second hypothesis is also supported.

We also conducted a spotlight analysis at one standard deviation below and above the mean of the independent self-index (M = 4.73) while holding the interdependent self-index (M = 4.17) constant at its mean [59]. The result showed that participants with a low level of independent self-construal did not differ in technology readiness toward the high-risk and low-risk mobile apps (β = 0.010, *p* > 0.1), whereas participants with a high level of independent self-construal perceived technology readiness more positively toward the low-risk mobile app than the high-risk mobile app (β = −0.619, *p* < 0.001) (see Figure 1a).

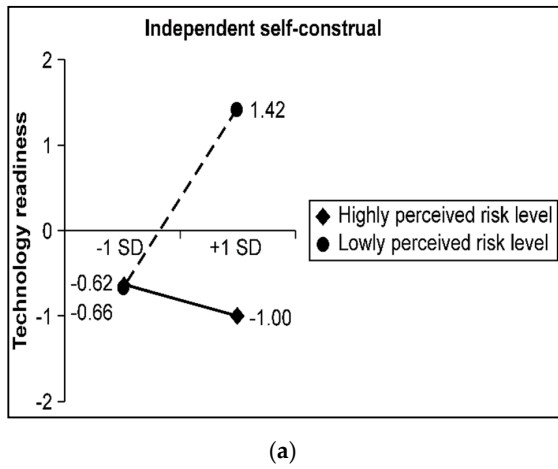

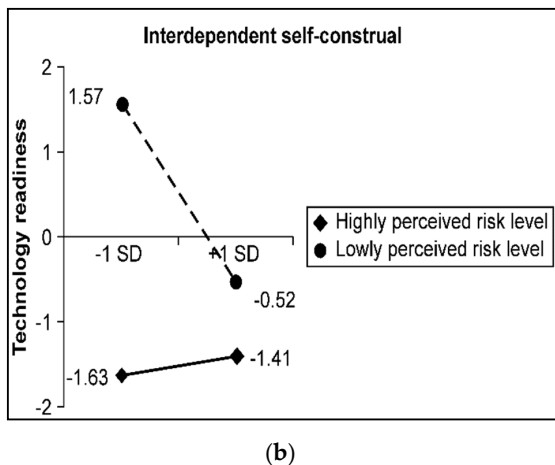

(**a**)  (**b**)

**Figure 1.** Technology readiness: (**a**) technology readiness toward the mobile app as a function of participants' independent self-construal and perceived risk; (**b**) technology readiness toward the mobile app as a function of participants' interdependent self-construal and perceived risk; SD is standard deviation.

We conducted a similar spotlight analysis at one standard deviation below and above the interdependent self-index (M = 4.17) while holding the independent self-index (M = 4.73) constant at its mean. The result showed that participants with a low level of interdependent self-construal perceived technology readiness more positively toward the low-risk mobile app than the high-risk mobile app (β = −0.715, *p* < 0.001). However, participants with a high level of interdependent self-construal did not differ in technology readiness toward the high-risk and low-risk mobile apps (β = −0.271, *p* > 0.1) (see Figure 1b).

### 4.3. Use Intention

We regressed the use intention on participants' mean-centered independent and interdependent self-indices; the perceived risk level; and the two interaction terms of perceived risk level × independent self and perceived risk level × interdependent self (see Table 5). The overall model was significant ($R^2$ = 0.144, F (5, 278) = 10.540, *p* < 0.001). The results showed the positive effect of the independent self-index (β = 0.417, *p* < 0.001) together with the negative effects of the interdependent self-index (β = −0.239, *p* < 0.01) and the perceived risk level (β = −0.216, *p* < 0.001) on use intention toward the mobile apps. More importantly, the interaction effects between the perceived risk level and the independent self-index (β = −0.266, *p* < 0.01), and between the perceived risk level and the interdependent self-index (β = 0.255, *p* < 0.01), were significant.

**Table 5.** Results of use intention regression analysis on different self-construals and perceived risk level.

| Model | Unstandardized Coefficients | | Standardized Coefficients | t | p |
|---|---|---|---|---|---|
| | Beta | Standard Error | Beta | | |
| (Constant) | 4.141 | 0.097 | | 42.492 | 0.000 |
| Independent self-construal (A) | 0.833 | 0.152 | 0.417 | 5.469 | 0.000 |
| Interdependent self-construal (B) | −0.294 | 0.096 | −0.239 | −3.056 | 0.002 |
| Perceived risk level (C) | −0.545 | 0.140 | −0.216 | −3.895 | 0.000 |
| A * C | −0.774 | 0.223 | −0.266 | −3.477 | 0.001 |
| B * C | 0.441 | 0.137 | 0.255 | 3.225 | 0.001 |

To understand the interaction effects, we examined the effects of the two self-construal indices on participants' use intention toward the mobile apps at each of the perceived risk levels. For the high perceived risk level, the independent self-index ($\beta = -0.003$, $p > 0.1$) was not significant whereas, for the low perceived risk level, the independent self-index was positive and significant ($\beta = 0.390$, $p < 0.01$). The interdependent self-index was not significant regardless of the perceived risk level ($p > 0.05$ for all).

We conducted a spotlight analysis at one standard deviation below and above the mean of the independent self-index (M = 4.73) while holding the interdependent self-index (M = 4.17) constant at its mean. The result showed that participants with a low level of independent self-construal did not distinguish between the high-risk and low risk mobile apps ($\beta = 0.016$, $p > 0.1$), whereas participants with a high level of independent self-construal perceived use intention more positively toward the low-risk mobile app than the high-risk mobile app ($\beta = -0.319$, $p < 0.05$) (see Figure 2a).

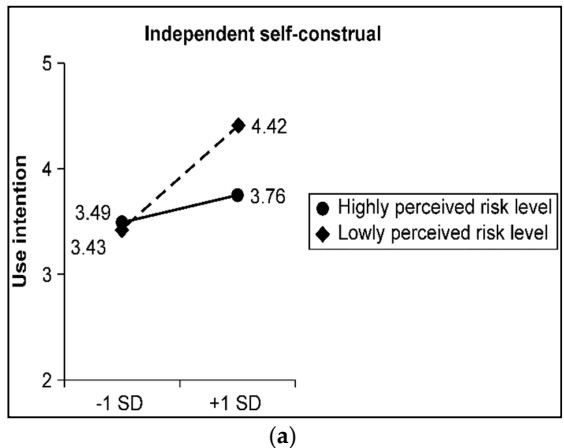 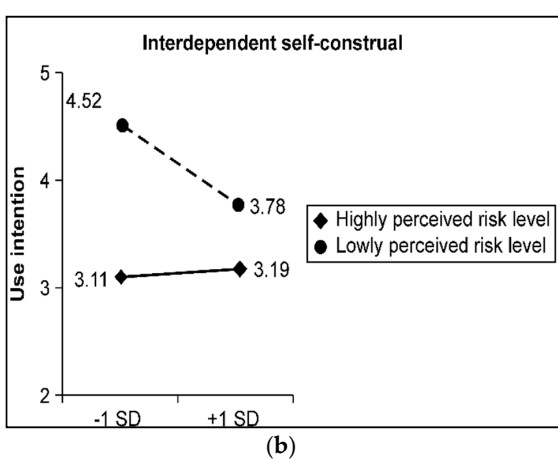

(a)                                                                    (b)

**Figure 2.** Use intention: (**a**) use intention toward the mobile app as a function of participants' independent self-construal and perceived risk; (**b**) use intention toward the mobile app as a function of participants' interdependent self-construal and perceived risk; SD is standard deviation.

We conducted a similar spotlight analysis at one standard deviation below and above the interdependent self-index (M = 4.17) while holding the independent self-index (M = 4.73) constant at its mean. The result showed that participants with a low level of interdependent self-construal had a more positive use intention toward the low-risk mobile app than the high-risk mobile app ($\beta = -0.569$, $p < 0.001$), whereas participants with a high level of interdependent self-construal did not differ in their use intention toward the high-risk and low-risk mobile apps ($\beta = -0.219$, $p > 0.1$) (see Figure 2b).

### 4.4. Mediation Analysis

To test our assumption that differences in use intention toward mobile apps stem from self-construal, we examined mediator models [60]. In this context, we expect that the effect of self-construal on use intention is mediated by technology readiness. Thus, we examined such mediation for the different conditions of perceived risk levels using bootstrapping methods (10,000 resamples, PROCESS macro model 4) [61].

A bootstrap analysis for the high-level perceived risk condition showed that the effects of independent and interdependent self-construal on use intention are not mediated by technology readiness, using a 95% confidence interval (CI) that includes 0 (for independent self-construal, CI = −0.2531 to 0.1232; for interdependent self-construal, CI = −0.0654 to 0.1463). The findings for the low-level perceived risk condition, however, showed that the mean indirect effects via technology readiness are significant. Specifically, the mean indirect effect via technology readiness on the relationship between independent self-construal and use intention is positive and significant (a × b = 0.30, 95% CI = 0.1415 to 0.5286) excluding 0. In addition, the mean indirect effect via technology readiness on the relationship between

interdependent self-construal and use intention is negative and significant (a × b = −0.19, CI = −0.3520 to −0.0859) excluding 0.

## 5. Discussion and Conclusions

Mobile apps are excellent information tools for tourism consumers. However, the greater the diversity of tourism information services, the greater the perceived risk. For example, tourism consumers could perceive the login and booking services of mobile apps as personal and financial risks. In this context, the technology readiness theory suggested by Parasuraman [9] proposed that, when people encounter new technologies, they focus not only on the positive aspects but also on the negative. Further, Parasuraman [9] argued that people who focus on the positive aspects of new technologies could use the new technologies, whereas those who focus on the negative aspects probably do not use them. Thus, tourism consumers with a high level of technology readiness could use tourism mobile apps. However, despite the burgeoning interest in technology readiness, research on its antecedents and their implications is sparse. In this study, we asserted that individual self-construal is the antecedent of technology readiness. We therefore examined the extent to which technology readiness differs across the concepts of self-construal (an individual variable) and perceived risk (a situational variable).

The findings showed that compared to consumers with interdependent self-construal, consumers with independent self-construal have a positive perception of technology readiness because they focus on what they gain from a mobile app. Consumers with interdependent self-construal focus on the losses they associate with a mobile app. More importantly, the results showed that the effect of self-construal on technology readiness is qualified by the perceived risk level. Specifically, the effect of self-construal on technology readiness is not significant in a high-level perceived risk condition because consumers' thoughts of losses are activated by a high perceived risk level regardless of their self-construal. In addition, the findings showed that the relationship between self-construal and use intention toward mobile apps is mediated by technology readiness only when perceived risk is at a low level.

The current research has theoretical implications for self-construal and technology readiness research. First, this study expands the boundaries of technology readiness research by demonstrating that self-construal can lead to technology readiness. In this regard, the study adopts a different approach to technology readiness by examining its antecedents rather than its consequences. The academic interest in the antecedents of technology readiness was insufficient, and the previous studies simply presented demographic variables as the antecedents. For example, Elliott et al. [62] suggested studying the antecedents of technology readiness by examining the effect of culture (e.g., American vs. Chinese) on technology readiness. Dutot [63] investigated the effects of consumers' age on technology readiness. By examining the role of an individual variable (self-construal), this research considers antecedents that have a more direct effect on technology readiness than those of prior studies.

This study also extends the prior research on self-construal and perceived risk by examining another research dimension. Aaker and Lee [29] suggested that individuals with an independent self-construal place relatively more weight on gains rather than losses, whereas individuals with an interdependent self-construal place relatively more weight on losses rather than gains. Further, Lin, Chang and Lin [52] examined the effect of message framing on product evaluation and showed that a high perceived risk level prompted a person to focus on losses no matter what type of individual self-construal applied. Our results show that these effects of self-construal and perceived risk apply similarly to research about the adoption of new technology.

In addition, this study has important practical implications for the tourism industry and mobile technologies. Tourism mobile apps can influence all stages of tourism consumers' decision-making [64], including the choice of destination, booking accommodation, and sharing travel experiences. This means that making mobile apps available for tourism

in specific countries can increase the number of tourism consumers who visit such countries. Indeed, many countries have recently developed tourism mobile apps to attract tourism consumers' interest. Examples are There's Nothing Like Australia, Visit Norway, Your Singapore Guide, and Visit Korea 3.0. In this regard, the study's findings show that tourism mobile apps should be developed, focusing on reducing tourism consumers' perceived risk. When the level of perceived risk is high, regardless of self-construal, consumers focus on the negative side of the technology. Based on the study's experimental stimulation, whether to approve the GPS function, the reliability of the payment system, and requests for personal data affect consumers' perceived risk. Therefore, it is necessary to consider ways to lower the perceived level of risk in relation to these functions. Furthermore, advertisements viewed by consumers through tourism mobile apps can be a good profit model for companies developing mobile apps; however, advertisements for companies and products that are unfamiliar to consumers can increase the perceived risk. Companies should develop tourism mobile apps after considering the awareness and image of a company or brand presented in advertisements can also affect consumers' perceived risk.

In addition, companies or countries that develop tourism mobile apps should implement differentiated strategies according to consumers' self-construal. Consumers are interested in different types of information according to their self-construal. If consumers searching for tourism-related information have independent self-construal, the tourism information presented to them through the tourism mobile app should be related to the benefits. For independents, it would be more effective to communicate the convenience and efficiency of the mobile app. However, if consumers have an interdependent self-construal, the information presented to them must be related to the loss. In other words, for them, it is necessary to suppress the induction of negative perceptions or emotions by increasing the reliability of the information obtained through the mobile app. At this time, information, such as other customers' reviews, can be an effective tool for lowering their perceived risk or uncertainty. To this end, companies should provide services that are discriminating in providing information after advance identification of consumers' self-construal.

An important limitation of this research relates to the stimulus materials and the measurement of perceived risk. As we presented a fictitious tourism mobile app to participants, tourism consumers' perceived risk could differ from actual situations, and although we measured perceived risk as overall risk, the type of risk that tourism consumers perceive could be different according to the functions of the mobile apps. For example, tourism consumers who use the booking services of mobile apps could be concerned about financial risk rather than other risk factors. Similarly, tourism consumers who use mobile apps to search for information about unfamiliar destinations could be concerned about performance and social risks rather than other risk factors. Future research could consider the weight of risk factors according to the different functions of real rather than fictitious tourism mobile apps.

**Author Contributions:** Conceptualization, J.C. and D.Y.; methodology, D.Y.; writing—original draft preparation, J.C.; writing—review and editing, D.Y.; All authors have read and agreed to the published version of the manuscript.

**Funding:** This research received no external funding.

**Institutional Review Board Statement:** Not applicable.

**Informed Consent Statement:** Informed consent was obtained from all subjects involved in the study.

**Data Availability Statement:** The authors confirm that the datasets analyzed during the study are available from the first author or the corresponding author upon reasonable request.

**Conflicts of Interest:** The authors declare no conflict of interest.

## Appendix A. Stimulus Materials

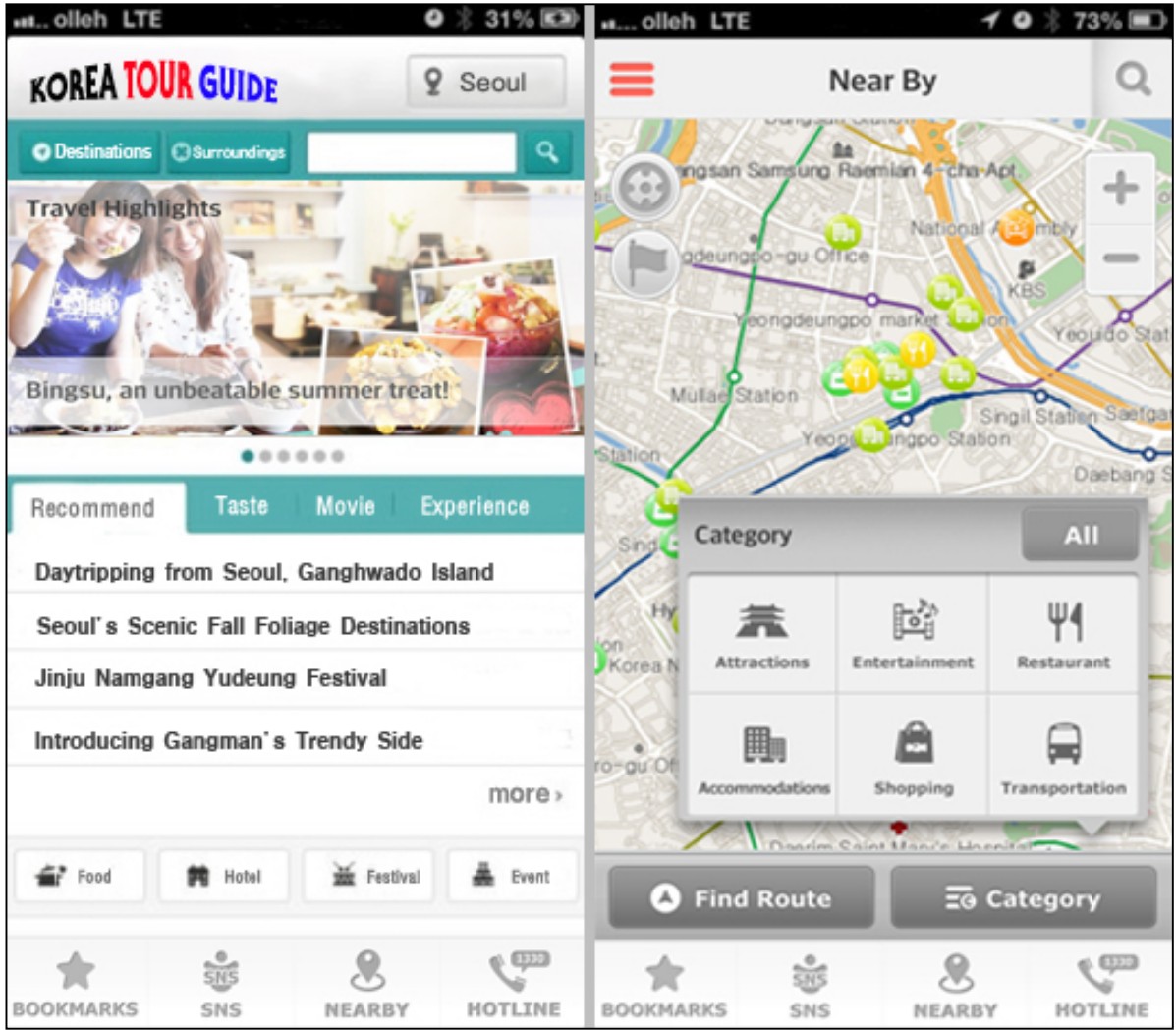

|  **High risk condition** | **Low risk condition** |
|---|---|

**High risk condition**

1. Name of mobile app: Korea Tour Guide
2. Providing and recommending tourist information by regions and themes
3. Providing navigation functions and local information by GPS (without GPS approval)
4. Functions that enable users to search for accommodation and make reservations organized by region (payment through mobile app)
5. Providing content about local tourism destinations
6. Uploading advertisements for unknown companies at the bottom of the mobile app
7. Not implementing security programs

**Low risk condition**

1. Name of mobile app: Korea Tour Guide
2. Providing and recommending tourist information by regions and themes
3. Providing navigation functions and local information by GPS (with GPS approval)
4. Functions that enable users to search for accommodation and make reservations organized by region
5. Providing content about local tourism destinations
6. Uploading advertisements for well-known companies at the bottom of the mobile app
7. Implementing security programs

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
