# Peer review of "The Impacts of Self-Construal and Perceived Risk on Technology Readiness"

_jtaer, doi:10.3390/jtaer16050089_

Round 1
Reviewer 1 Report
Dear authors, I want to congratulate you on the development of your article. The study is new and interesting but for publication I recommend that you make changes.
- The theoretical framework is limited and insufficient. The literature review should be improved and enlarge.
- It is advisable to separate the introduction from the theoretical framework.
- They must explain and justify the methodology used, it is very important.
- They must better classify the sample and justify it.
- They must detail and justify what statistical software they use.
- It would be convenient to broaden the managerial implications.
- They must justify the scale of measurement used.
- They must update the bibliographic references.
- I would also urge the authors to read the articles listed below before completing the manuscript revision and make citations to them to strengthen the manuscript's theoretical background.
Grewal, D., Hulland, J., Kopalle, PK y Karahanna, E. (2020). El futuro de la tecnología y el marketing: una perspectiva multidisciplinar.
Martínez-Navalón, J. G., Gelashvili, V., & Saura, J. R. (2020). The Impact of Environmental Social Media Publications on User Satisfaction with and Trust in Tourism Businesses. International Journal of Environmental Research and Public Health, 17(15), 5417.
Ji, X., Huang, K., Jin, L., Tang, H., Liu, C., Zhong, Z., ... y Yi, M. (2018). Descripción general de la tecnología de seguridad 5G. Science China Information Sciences , 61 (8), 1-25.
Reviewer 2 Report
The paper is generally well written and it approaches an interesting and meaningful topic. However, before it can be accepted for publishing, there are a few things worth mentioning, with the aim to bring a little overall improvement.
First, in the Introduction and Conclusion section, the authors should further underline the novelty of the conducted research in comparison to what other researchers wrote on related topics. Then, although these issues are mentioned, still, a little more emphasize on the present and future practical implications and potential benefits of the findings would be of help if it were more detailed, especially in the Abstract section. Somehow, the reader finds rather hard to comprehend the whole point of the research unless going over the whole paper.
Another thing is related to line 171, paragraph 2.2. Research Procedure. Here things should also be developed a little more, with more details regarding when and how the sample of 284 participants was recruited and what makes them to be a relevant sample for the conducted study (for instance, what was their background etc.). Then, a little more details about the mobile app that was used (as it is presented in the Appendix), even if it was a fictitious one, for instance: how was it written/developed, what software/platform was used, if it was developed by the authors or by a third party etc. and how was it deployed to all the 284 participants? All these issues will provide more clarity and depth from a scientific perspective and in the same time will better cover the limitations of the conducted research.
Round 2
Reviewer 1 Report
I want to congratulate you on the development of your article.